# Transcriptional Control of Subcutaneous Adipose Tissue by the Transcription Factor CTCF Modulates Heterogeneity in Fat Distribution in Women

**DOI:** 10.3390/cells13010086

**Published:** 2023-12-30

**Authors:** Edina Erdos, Katalin Sandor, Crystal L. Young-Erdos, Laszlo Halasz, Steven R. Smith, Timothy F. Osborne, Adeline Divoux

**Affiliations:** 1Division of Diabetes Endocrinology and Metabolism, Departments of Medicine, Biological Chemistry and Pediatrics, Johns Hopkins University School of Medicine, Institute for Fundamental Biomedical Research, Johns Hopkins All Children’s Hospital, St. Petersburg, FL 33701, USA; 2Department of Chemistry & Biochemistry, Eckerd College, St. Petersburg, FL 33711, USA; 3Translational Research Institute, Adventhealth, Orlando, FL 32804, USA

**Keywords:** fat distribution, adipose-derived stem cells, CTCF, RNAPII, bivalent chromatin, epigenetic, METRNL

## Abstract

Determining the mechanism driving body fat distribution will provide insights into obesity-related health risks. We used functional genomics tools to profile the epigenomic landscape to help infer the differential transcriptional potential of apple- and pear-shaped women’s subcutaneous adipose-derived stem cells (ADSCs). We found that CCCTC-binding factor (CTCF) expression and its chromatin binding were increased in ADSCs from pear donors compared to those from apple donors. Interestingly, the pear enriched CTCF binding sites were located predominantly at the active transcription start sites (TSSs) of genes with active histone marks and YY1 motifs and were also associated with pear enriched RNAPII binding. In contrast, apple enriched CTCF binding sites were mainly found at intergenic regions and when identified at TSS, they were enriched with the bivalent chromatin signatures. Altogether, we provide evidence that CTCF plays an important role in differential regulation of subcutaneous ADSCs gene expression and may influence the development of apple vs. pear body shape.

## 1. Introduction

Although obesity (BMI > 25) is associated with a higher prevalence of the metabolic syndrome, fat distribution is a more accurate predictor for the risk of developing cardiovascular diseases and type 2 diabetes (T2D) [1]. Correlation studies have shown that excess upper body adipose tissue accumulation, including both abdominal (ABD) subcutaneous and visceral depots, are associated with hypertension, dyslipidemia and insulin resistance [2]. Conversely, the preferential accumulation of excess fat in the lower body subcutaneous fat depots (femoral, gluteal, and gluteofemoral (GF)) has been shown to be protective against the same risk factors even when corrected for BMI [3,4]. The specific role of subcutaneous adipose tissue (SAT) in the development of metabolic syndrome is not yet fully understood. One hypothesis proposes that inadequate SAT expansion during periods of energy excess could lead to ectopic lipid accumulation in non-lipid storage organs (liver, pancreas, muscle), resulting in cellular dysfunction [5]. According to this hypothesis, in some women, ABD and GF depots do not have the same capacity for expansion, leading to an apple body shape with increased risk for developing metabolic complications. To this end, understanding why some women preferentially store excess fat in their ABD fat depot (apple-shaped) while others deposit excess fat in their GF depot (pear-shaped) and are protected from metabolic disease is an important clinical and mechanistic question.

In addition to intrinsic defects in adipocyte function and variations in hormonal responses, genetic and epigenetic factors contribute to body fat patterning [6,7,8]. Notably, our lab and others previously identified in ABD and GF adipose tissues specific transcriptional signatures associated with body shape along with epigenetic patterns [9,10] that were partially maintained in adipose-derived stem cells (ADSCs) expanded in culture [9,11]. These preliminary data suggest that key differential epigenetic programs could be identified by examining the ADSCs collected from ABD and GF depots in subjects of apple vs. pear shapes. In addition, epigenetic mechanisms are known to control cellular differentiation, including adipogenesis [12,13], through regulation of gene expression to (i) maintain a stem cell population and (ii) activate cellular differentiation.

CCCTC-binding factor (CTCF) is a ubiquitous transcription factor that binds DNA to control the spatial organization of chromatin and modulate transcription. CTCF both positively and negatively regulates gene expression in a gene-specific and context-specific manner [14,15,16]. Importantly, the sequence-specific binding of CTCF is variable and is impacted by partner protein association and the nearby epigenetic context. In adipose tissue, CTCF is essential to adipogenesis but neither the underlying mechanism nor whether it influences body shape-selective gene expression has been reported [17,18].

In the current study, we examined the role of CTCF in human ABD and GF ADSC biology and its role in body-shape specific transcriptome regulation. We used ChIP-seq to compare the binding of CTCF in ABD and GF ADSCs in volunteers that represent the extremes of apple vs. pear body fat distribution. We further used a combination of ChIP-seq assays, targeting key histone modifications (H3K4me3, H3K27me3, H3K4me2, H3K27ac, and H3K9me3) along with ATAC-seq to probe the chromatin state near the body shape-specific CTCF binding sites. These signatures were combined with RNA polymerase II ChIP-seq and RNA-seq to probe the transcriptional mechanisms contributing to differential gene activity and to specifically study the impact of CTCF binding on body shape-specific gene regulation.

## 2. Materials and Methods

Participants, tissue collection, and isolation of human adipose-derived stem cells.

The method of recruitment, and the clinical and biochemical parameters of the subjects are presented in Divoux A. et al. (2020) [19]. All procedures were performed under a research protocol approved by the AdventHealth Institutional Review Board. A subgroup of nine healthy premenopausal, weight-stable women were used for this study (age = 34 ± 9.6 years; BMI = 29.2 ± 2.26 kg/m^2^). Briefly, paired abdominal (ABD) and gluteofemoral (GF) SAT samples were obtained from each participant, and the stromal-vascular fractions (SVF) were isolated by collagenase digestion and centrifugation from both AT depots. To obtain the adipose-derived stem cells (ADSCs), we plated the SVF in αMEM supplemented with 10% FBS for 24 h. After this first incubation, the non-adherent cells were washed with PBS, and the adherent cells were grown in a proliferation medium containing 2.5% FBS, FGF, and EGF. Endothelial cells were removed from the cultured ADSCs using magnetic beads (StemCell Technologies, Vancouver, BC, Canada) as described in (Divoux 2021) [9]. The ADSCs were plated, grown, and differentiated as previously described [20] to study CTCF expression during differentiation. A mid-cocktail of differentiation (with only IBMX, dexamethasone, rosiglitazone, and insulin) was used as described in (Zaragosi et al., 2010) [21] to obtain cells with different levels of differentiation. The cells were harvested across differentiation (Day 0–12) or only before and at the end of differentiation (Day 0 and Day 12), RNA was extracted, and target genes were measured by real-time PCR using a ViiA7 sequence detection system (Life Technologies, Carlsbad, CA, USA) and Taqman technology suitable for relative gene expression quantification using the following parameters: one cycle of 95 °C for 10 min, followed by 40 cycles at 95 °C for 10 s and 60 °C for 1 min.

Chromatin immunoprecipitations

Chromatin immunoprecipitations (ChIPs) were performed on confluent ABD and GF ADSCs as described [22]. The following ChIP grade antibodies were used: rabbit anti-H3K4me3 (Diagenode—C15410003), rabbit anti-H3K4me2 (DIagenode—pAb-035-050), rabbit anti-H3K27me3 (Diagenode—C15410069), rabbit anti-H3K27Ac (Abcam—ab4729), anti-CTCF antibody (Actif Motif—61311), and rabbit anti-RNAPII (Abcam—ab5095) to study RNA Polymerase II binding.

ChIP-qPCR

Chromatin immunoprecipitations (ChIPs) were performed and analyzed as described in (Divoux et al., 2018) [11]. Briefly, cells were crosslinked with DSG and 1% formaldehyde and quenched with glycine. The rabbit monoclonal YY1 (Cell Signaling—D5D9Z) antibody was used to isolate YY1; a rabbit IgG was used as a negative control. Chromatin–antibody complexes were immunoprecipitated with dynabeads (Life Technologies) blocked with PBS 1% BSA. ChIP-qPCR assays were performed using the primers listed in Appendix A. Primers were designed to amplify the promoter regions from body shape enriched CTCF binding sites, a region of the NDUFA6 locus for a positive control and GAPDH for a negative control.

Assay for transposase-accessible chromatin (ATAC)

ATAC was performed as previously described by Divoux A. et al. [11] on ABD and GF ADSCs. Briefly, isolated nuclei were tagmented using a Nextera DNA library preparation kit (Illumina, San Diego, CA, USA), and the resulting DNA was purified and amplified with a Kapa HiFi Hot Start Kit (Kapa Biosystems, Wilmington, MA, USA). The purified libraries were sequenced on a HiSeq 2500.

Sequencing libraries preparation

RNA-seq, ATAC-seq, and ChIP-seq libraries were prepared and sequenced using standard Illumina protocols for a HiSeq 2500 instrument.

RNA sequencing and analysis

RNA sequencing was performed as described [11], using genome hg19 as the reference. Aligned RNA-seq data (bam files) were used for further analysis via HOMER (Hypergeometric Optimization of Motif EnRichment) bioinformatic program tools. Firstly, tag directories were created with the makeTagDirectory command. Raw count files were generated with the analyzeRepeat.pl command rna, and raw options were then handled by the getDiffExpression.pl command using DESeq2, an R package. The outputs contained rlog variance stabilized values too.

ChIP-seq and ATAC-seq analysis

ChIP-seq and ATAC-seq analysis were performed as described in (Erdos et al., 2022) [23].

Differentially binding analysis

Aligned ChIP-seq data (bam files) were used for further analysis via HOMER (Hypergeometric Optimization of Motif EnRichment) bioinformatic program tools. First, tag directories were created and combined with the makeTagDirectory command. Then, peak files were generated with the findPeaks.pl command from a combined tag directory (combined all six subjects). Differentially CTCF binding sites were found with the getDiffExpression.pl command using DESeq2, an R package. To get differential binding at TSS (on the ±2 kb flanking regions), the annotatePeaks.pl command was used with the tss option. and the given raw count files were used to the getDiffExpression.pl command using DESeq2, an R package. The outputs of the getDiffExpression.pl command contained rlog variance stabilized values too.

Motif analysis

Motif enrichment analysis was carried out by HOMER (Hypergeometric Optimization of Motif EnRichment) with the findMotifs.pl command. It was performed on the −1000 bp and +250 bp flanking regions of the TSS. Motif densities were calculated by HOMER with the annotatePeaks.pl command.

ChromHMM

Chromatin states were learned by applying the ChromHMM (version 1.21) hidden Markov model (HMM) algorithm at a 200 bp resolution to seven data tracks (RNAPII, CTCF, ATAC, H3K27ac, H3K27me3, H3K4me2, and H3K4me3) from each of the three ABD ADSC and three GF ADSC samples in each group (apple and pear).

Visualization and statistics

To plot heatmap of ChIP-seq signals and histograms of ChIP-seq signals, we used the plotHeatmap and plotProfile commands within deepTools (version 3.5.1) (Ramírez et al., 2014) [24]. Volcano plots were created using the R package EnhancedVolcano [25]. To plot heatmaps of gene expression, we used the ComplexHeatmap package [26]. ChromHMM heatmaps were generated using the EnrichedHeatmap [27]. The Corrplot R package was used to visualize the heatmap for correlation (Spearman correlation).

Boxplots, bar graphs, histograms, and line plots were created using the ggplot2 and ggpubr package. To visualize the intersections, a proportional Venn diagram was used (DeepVenn).

To determine statistical significance, ggpubr was used. To compare two independent groups, we used the unpaired two-samples Wilcoxon test (Wilcoxon rank sum test). To compare paired data, we used the paired samples Wilcoxon test (Wilcoxon signed-rank test). Spearman rank was used for correlation calculation.

Data reposition.

All sequencing data have been deposited to the NCBI GEO database (http://www.ncbi.nlm.nih.gov/geo/ (accessed on 30 June 2023)) under accession numbers GSE224770 (ChIP-seq and ATAC-seq), GSE193812 (RNA-seq from 18 ADSCs), and GSE143450 (RNA-seq from adipocytes). GTEx data were obtained from pht002742.v9.p2 version.

## 3. Results

### 3.1. Leg Fat Mass Is Associated with Reduced Visceral Adipose Tissue and Healthier Clinical Profile in Premenopausal Women

Historically, the waist-to-hip ratio (WHR) has been a convenient and simple index to monitor the partitioning of adipose tissue in the upper vs. lower body areas. Its clinical utility has been limited because of assessment error during waist and hip measurement, and waist circumference does not distinguish between SAT vs. visceral adipose tissue (VAT). A more precise evaluation of fat distribution is achieved with dual X-ray absorptiometry (DEXA), which accurately measures the composition and distribution of total body fat and lipids contained in all body areas, including the abdomen and gluteal, femoral, and distal leg. In our study, we collected clinical variables along with WHR and fat distribution DEXA scans performed on nine relatively healthy premenopausal women with a range of body fat distribution (BMI = 28.6 ± 2.68 kg/m^−2^ and age = 35.4 ± 8.34 years). A correlation matrix was generated to provide insight into the interrelationships of these variables with WHR, VAT mass, and leg fat accumulation (LFM, calculated as the percentage of fat accumulated in the legs relative to total fat mass). LFM was negatively correlated with MRI-measured liver fat, VAT mass, and circulating neutrophils and positively correlated with circulating lymphocytes (Figure 1A). Consistent with the measurement error, there were fewer significant correlations between WHR and the clinical parameters measured during our investigation (Figure 1A). These results illustrate that LFM provides a more informative measure of metabolic risk traits than WHR, notably through its high negative association with VAT mass.

Next, we clustered the data from the nine women into bins of three based on LFM (Figure 1B upper left graph). This showed there was a significant variation in clinical parameters, including VAT, liver fat content, and circulating triglycerides (TG), only when the two extreme bins were compared (Figure 1B). Thus, for our cell molecular investigations, we decided to focus on the two extreme groups with the lowest and highest LFM and VAT mass, respectively. We refer to these two groups as pear (highest LFM and lowest VAT) and apple (lowest LFM and highest VAT).

From these six women, we isolated and cultured ABD and GF ADSCs to compare their transcriptional potential (both RNA-seq and RNAPII ChIP-seq) and epigenomic landscape (ChIP-seq for key histone modifications and CTCF along with ATAC-seq). The experimental design of our study is summarized in Figure 1C. We also leveraged data previously generated in our lab on these same subjects where we isolated adipocytes and analyzed their transcriptomes by RNA-seq [19].

### 3.2. CTCF Is a Key Factor Involved in Body Shape-Specific Transcriptional Regulation in Human ADSCs

To determine the transcriptional profile of apple vs. pear ADSCs, we first analyzed RNAPII ChIP-seq data from paired ABD and GF ADSC chromatin with an antibody directed to the elongating form of RNAPII (Pol II-phosphoSer2) [28]. To focus on shape-specific active gene activation, we compared the RNAPII localization at gene transcriptional start sites (TSSs) between apple and pear samples. In the ABD depot, 2766 and 1084 genes displayed higher RNAPII binding in pear and apple cells, respectively (Figure 2A), whereas 1788 total genes displayed body shape-enriched RNAPII binding in GF ADSCs (945 enriched in pear and 843 enriched in apple; Appendix A). We observed that the gene regions of several transcription factors showed higher RNAPII binding in pear ABD ADSCs compared to that in apple ABD ADSCs. This included *ATF4*, *CTCF*, *E2F4*, and *YY1* along with the TGF-β ligand *GDF5* (Figure 2A). IGV screenshots for pear enriched RNAPII binding at the TSSs of *CTCF*, *YY1*, and *ATF4* are shown in Figure 2B. In both groups, Pol II-phosphoSer2 predominantly occupied the promoter regions and peaked around the TSS (Figure 2C in the ABD depot and Appendix A in the GF depot). However, metagene profiles showed body shape-selective RNAPII binding throughout whole gene bodies and not just at the TSS consistent with differential gene transcription between apple and pear ADSCs (Figure 2C in the ABD depot and Appendix A in the GF depot). To confirm this finding, we intersected the list of genes with body enriched RNAPII binding sites at their TSS with the list of differentially expressed genes identified by RNA-seq analysis. We found only a low correlation between the number of genes detected as significantly differentially expressed by both methods (*p* < 0.05, *n* = 246 in ABD ADSCs, colored dots in Figure 2D; *n* = 38 in GF ADSCs, colored dots in Appendix A). This suggests that the mechanisms that result in ADSC differential gene expression likely involve a combination of epigenetic and other transcriptional effects that influence, for example, transcriptional initiation, RNAPII pausing, and the post-transcriptional process.

Importantly, CTCF was identified as a pear enriched mRNA in ABD ADSCs by RNA-seq and RNAPII binding analysis, implicating CTCF in the regulation of body shape-specific gene expression. Knowing that CTCF is essential for adipogenesis [17,18], we next explored CTCF expression during in vitro differentiation of human ABD ADSCs. Using a standard differentiation cocktail for inducing adipogenesis that results in almost 100% conversion of the ADSCs, we observed that total CTCF RNA expression was significantly reduced rapidly (~3-fold lower) and it remained low across the differentiation time course (Figure 2E). We next used a moderate cocktail in vitro that did not force all the ADSCs to differentiate (see Material and Methods) and allowed us to obtain differential levels of overall differentiation that reflected the adipogenic potential of the initial donors. Interestingly, we found a negative correlation between the level of expression of *CTCF* at the end of differentiation (D15) and the level of expression of key marker genes that reflect the maturity of adipocytes (*CEBPA*, *PPARG*, *FABP4*, and *FASN*; Figure 2F). The observed inverse correlations are consistent with prior studies [17,18] and suggest that CTCF suppresses these key adipogenic genes.

### 3.3. Differential CTCF Enrichment in Apple vs. Pear ADSCs

CTCF is a chromatin regulator with highly versatile functions, including direct gene activation and repression, as well as more broad roles in chromosome insulation and imprinting. To further investigate the role of CTCF in ADSC biology, we evaluated its potential role in body shape-specific transcriptional regulation by comparing genome-wide CTCF occupancy in apple vs. pear ADSCs isolated from ABD and GF adipose tissue. Among the over 89,000 total CTCF binding sites identified in ABD ADSCs, 14,649 were body shape-enriched (*p* < 0.05; Figure 3A). Interestingly, 28% of the 7306 pear enriched CTCF sites were localized at the TSSs of genes compared to only 2% of the 7343 apple enriched sites being localized to gene TSSs (Figure 3B, in brown). Conversely, 57% of the apple enriched CTCF binding sites were found in the intergenic regions compared to only 22% for the pear enriched sites (Figure 3B, in purple). These differences were even more pronounced in GF ADSCs, where 40% of the pear enriched sites were found at the TSSs of genes, compared to only 1.8% of the apple enriched sites (Appendix A).

We next explored whether CTCF binding was associated with other regulatory elements that might reveal context dependent roles for CTCF in different ADSC populations. We examined the chromatin landscape around the body shape-specific CTCF binding sites with ChIP-seq data from these same six subjects. We combined our ChIP-seq data for H3K4me3, H3K4me2, H3K27me3, H3K27ac, and RNAPII along with ATAC-seq results using the computational integrating ChromHMM package. This analysis revealed 10 combinatorial chromatin emission states as listed in Figure 3C for ABD samples and in Appendix A for GF samples. The ChromHMM signatures around the body shape-enriched CTCF binding sites revealed a striking difference for the chromatin neighborhood the apple vs. pear CTCF binding regions (Figure 3D in the ABD depot and Appendix A in the GF depot). The apple enriched CTCF binding sites were predominantly found alone without any other accompanying histone marks (note the light green color in the left two panels). In contrast, most of the pear enriched CTCF binding sites were co-localized with RNAPII and open chromatin (ATAC-seq peak) along with all the active histone marks that were analyzed (note the red color in the right two panels). This emission state is defined as “Active Tss” (labelled TssA in Figure 3C). These two distinct patterns suggest a different role for CTCF in body shape-specific ADSCs. Firstly, a likely structural role mediated predominantly by CTCF alone at the apple enriched CTCF sites. Secondly, a role in transcription activation as a part of a multi-factor complex at the pear enriched CTCF sites. Despite the dramatic difference between the apple and pear enriched CTCF binding regions, the ChromHMM patterns around the body shape enriched CTCF binding sites were the same in apple and pear samples (Figure 3D and Appendix A), indicating that body shape-preferential CTCF binding is not associated with a drastic overall chromatin modification between apples and pears.

### 3.4. Enrichment of CTCF at Active Transcription Start Site Containing YY1 Motifs Selectively in Pear ADSCs

When we focused on the chromatin landscape around the body shape-enriched CTCF binding sites at gene promoter regions (+/−2000 bp around the gene TSS), we found between seven and eight-fold fewer CTCF binding sites enriched in apple TSSs than in pear TSSs (690 vs. 4791 in ABD ADSCs, Figure 4A,B; 245 vs. 1970 in GF ADSCs, Appendix A). Interestingly, a high proportion of genes with apple selective CTCF binding were associated with both the repressed state enriched for H3K27me3 (grey in Figure 4A and in Appendix A) and a recognized bivalent chromatin state containing H3K27me3 along with at least one activating mark (“TssBiv”, pink in Figure 4A and in Appendix A). Conversely and as expected from the analysis in Figure 3D, almost all the pear enriched CTCF binding sites were associated with an active TSS chromatin state (red in Figure 4B for ABD depot and in Appendix A for GF depot).

The Venn diagrams in Figure 4C,D show that in ABD ADSCs, 28% of the apple enriched CTCF binding at gene promoters was indeed associated with a bivalent chromatin state, compared to only 4% for the pear enriched CTCF binding sites. Similar results were found in GF ADSCs (Appendix A). It has been proposed that bivalent states are “poised” and could easily be converted to either active or repressed states by selective removal of the repressing or activating mark, respectively [29]. We reasoned that CTCF might mark genes with the bivalent ChromHMM state that are activated or repressed during adipocyte differentiation. To explore this hypothesis, we compared the level of expression of the genes with bivalent states at their TSS between ADSCs and their matched isolated adipocytes. The box plot comparisons in Figure 4E,F show that the bivalent marked genes were expressed at higher levels in adipocytes (positive fold change, purple box plot), which is consistent with the bivalent marked genes being maintained in a “poised” state in ADSCs that are then converted to the active state upon differentiation. As suggested by our earlier data in vitro (Figure 2E,F), the non-bivalent marked genes were expressed at higher levels in ADSCs compared to those in adipocytes (yellow boxplot Figure 4E,F).

CTCF is a chromatin factor with multiple context-dependent functions [30] that rely on interactions with different collaborating proteins including, but not limited to, other transcription factors, transcriptional cofactors, RNA polymerase II, and subunits of the cohesin complex [31,32]. The different ChromHMM signatures for the apple vs. pear enriched CTCF discussed above suggest that there are potential co-factors interacting with CTCF in ADSCs to regulate pear enriched genes. We performed a motif analysis for the body shape-specific CTCF binding sites located at the gene TSSs. This analysis revealed an enrichment for the expected CTCF motif in apple (Figure 4G) consistent with the ChromHMM signature analysis in Figure 3D (two left panels). Unexpectedly, the most enriched motif in the pear-specific CTCF TSS binding sites was YY1 (Figure 4H). This YY1 motif was also observed in pear-specific CTCF binding sites at gene TSSs in the GF depot (Appendix A). We used a sliding scale motif distribution analysis to specifically evaluate the enrichment for CTCF and YY1 motifs +/−1000 bp around the TSSs of the genes with body shape-specific CTCF binding sites (Figure 4I,J). This analysis showed that there was no significantly preferred location for the promoter proximal CTCF motif or YY1 motif in the apple-specific CTCF TSSs (Figure 4I). In contrast, there was a significantly preferred location of the YY1 motif at the TSSs of the genes with pear-specific CTCF binding sites (Figure 4J).

YY1 has been shown to directly interact with CTCF [33], and the selective prevalence of YY1 motifs at the TSSs of pear enriched CTCF suggests that YY1 selectively influences pear gene expression through interacting with CTCF.

We next compared the changes in transcriptional activity of genes with CTCF vs. YY1 motifs at gene TSS by using RNA-seq and RNAPII ChIP-seq data in the three pear subjects (Figure 4K). The genes with YY1 motifs (blue) showed higher RNA expression levels and RNAPII binding compared to those of the genes with CTCF motifs alone (yellow). Altogether, these results suggest CTCF may play different roles in gene expression regulation in apple vs. pear ADSCs according to the presence or absence of the YY1 motif within the gene TSS.

Finally, to determine if YY1 binds to the pear-specific CTCF binding sites containing the YY1 motif, we performed a YY1 ChIP-qPCR for the promoters of several of these genes using chromatin from both apple and pear ABD ADSCs. Consistent with our prediction, ChIP-qPCR showed YY1 enrichment at the TSS of several of the predicted genes (Figure 4L). Importantly, however, there was no difference for YY1 binding in apple vs. pear subjects, suggesting that depot-specific effects of CTCF are not simply due to differential binding of the pear-selective co-factor YY,1 and other CTCF partners may be important.

### 3.5. Coordinated CTCF and RNAPII Binding at TSS of Genes Correlates with Body Shape-Specific Gene Expression

We decided to investigate further the role of CTCF in body shape gene expression regulation by comparing CTCF and RNAPII ChIP-seq data at the gene promoter. The similar heat map profiles between CTCF and RNAPII binding around the TSS of genes with pear enriched CTCF sites suggest that CTCF binding occurs alongside the binding of RNAPII (Figure 5A for ABD samples and Appendix A for GF samples). This observation is consistent with the results of our ChromHMM analysis in Figure 4B and Appendix A and with previous reports [34]. A graphic genome browser view showing this association for a select group of apple and pear enriched genes is depicted in Figure 5B. The quantification of RNAPII and CTCF peak read densities around the TSS confirmed that CTCF and RNAPII occupancy were higher in apple compared to that in pear samples at the apple enriched CTCF binding genes (Figure 5C for ABD and Appendix A for GF). Similarly, CTCF and RNAPII occupancy was higher in pear samples compared to that in apple samples at pear enriched CTCF binding genes (Figure 5D and Appendix A). However, a direct gene-by-gene comparison revealed that only a small proportion of the total apple enriched CTCF binding genes were also preferentially enriched for RNAPII in the apple samples in ABD ADSCs and in both groups in GF ADSCs (Venn diagram in Figure 5E for ABD and in Appendix A for GF). In contrast, 45% of the genes with pear enriched CTCF binding sites at their TSSs also contained higher RNAPII density in pear samples (Venn Diagram Figure 5F). This pattern suggests that CTCF and RNAPII function in concert to influence body shape-specific gene expression preferentially in pear ADSCs.

To validate this hypothesis and further explore the implication for CTCF/RNAPII co-occupancy in ADSC gene regulation, we evaluated genes with overlapping CTCF and RNAPII binding at their promoter (overlap in Figure 5E,F) using RNAPII binding analysis. The heatmaps in Figure 5G represent the RNAPII binding data in the three apple and three pear ADSCs around the promoters of the 183 and 2248 genes with apple vs. pear CTCF/RNAPII enriched binding sites, respectively. Importantly, the data for the six subjects was plotted relative to their level of LFM. We observed a high segregation of gene expression between apple (low LFM) and pear (high LFM) samples for almost all of the genes analyzed. We further quantified the correlation between LFM and RNAPII binding by attributing a scaled value for RNAPII binding intensities for each subject (Figure 5H). We found 25% of the genes with apple enriched CTCF and RNAPII binding sites at their promoter negatively correlated with LFM (upper graph), while more than 65% of the genes with pear enriched CTCF and RNAPII binding sites at the promoter positively correlated with LFM (lower graph). Similar results were found in the GF depot (Appendix A); however, a lower percentage of genes were correlated with LFM (Appendix A). Interestingly, two genes, *NRG1* and *BHMT,* with apple enriched CTCF binding sites at their promoter, were also expressed at lower levels in apple compared to pear samples (Figure 5H). NRG1 serves as regulator of adipogenic differentiation in human ADSCs [35], and BHMT promotes adipocyte commitment and adipose tissue expansion [36]. ChromHMM representation at the promoters of these two genes revealed the presence of the bivalent state at their TSSs (Appendix A).

### 3.6. METRNL: A New CTCF Target Gene Associated with Differential Fat Distribution in Women

The data presented so far suggest that in ADSCs, CTCF, YY1, and RNAPII cooperate to selectively target TSSs to influence gene expression. To confirm these observations on a larger group of subjects, we used RNA-seq data generated in ABD ADSCs isolated from 18 women with a differential body shape. From the list of genes with body shape-enriched CTCF and RNAPII binding sites at their promoter, 19 were positively or negatively correlated with LFM (Figure 6A, heatmap representation). Among these genes, *AGPAT2*, *ARL5C*, and *METRNL* have previously been shown to be involved in adipose tissue biology [37]. *METRNL* expression was found to be negatively correlated with LFM (Figure 6B), and interestingly, its expression was also found to be positively correlated with the known downstream cardiovascular disease (CVD) risk factors (VAT mass, VLDL, and circulating TG; Figure 6C).

We confirmed a parallel increase in both CTCF binding and LFM at the *METRNL* locus within the CTCF binding data from the six subjects used for our chromatin analyses (Figure 6D). These results indicated that simultaneous CTCF and RNAPII binding is not associated with an activation of transcription. Interestingly, ChromHMM representation showed the presence of bivalent TSS marks at the *METRNL* promoter (Figure 6D lower part, blue segment), suggesting that the *METRNL* gene is “poised” in ADSCs. *METRNL* has previously been shown to activate PPARG pathways [38], so we looked at the correlation between *METRNL* and *PPARG* gene expression in ADSCs isolated from the larger 18-subject group. Interestingly, we observed a positive correlation between the expressions of these two genes, but only in pear subjects.

Finally, to evaluate these correlations within a larger cohort, we used RNA-seq data from human adipose tissue derived from GTEx datasets (*n* = 163 women). From our list of 19 genes, expression of 16 (including *METRNL*) showed a positive correlation with *CTCF* expression in this larger data set (Figure 6E). *CTCF* expression also showed a strong correlation with *YY1* in this analysis, in favor of an interaction between CTCF and YY1 as suggested by our data in Figure 4 and prior literature [39].

## 4. Discussion

CTCF is a ubiquitous chromatin factor that has been previously described as a potential molecular regulator of pre-adipocyte differentiation [18]. In this work, we studied for the first time in human subcutaneous adipose progenitor cells (or ADSCs) genome-wide binding of CTCF, in parallel with an in-depth evaluation of the neighboring epigenomic landscape. By comparing the results observed in ADSCs isolated from apple-shaped women to those observed in ADSCs isolated from pear-shaped women, we found that half of the apple enriched CTCF binding sites were observed in intergenic regions. This observation is consistent with a previous report describing the role of CTCF in other cells [40]. This was different in ADSCs from pear-shaped donors where a high percentage of pear shape enriched CTCF binding sites were found within gene promoter regions. Interestingly, these regions were also enriched for RNAPII as well as key histone modifications known to mark regions of active transcription. Our data also suggest that YY1 may play a key role in driving pear enriched CTCF binding at the TSSs of specific genes. Interestingly, downregulation of YY1 has been associated with increased expression of CHOP-10, an inhibitor of adipocyte differentiation [41,42]. The potential interaction between CTCF and YY1 that we highlighted in pear ADSCs could promote adipogenesis specifically in these progenitor cells. In contrast, apple enriched CTCF sites were not highly associated with other epigenomic marks or the YY1 motif. Altogether, these results suggest an active role of CTCF in body shape-selective ADSC gene regulation and potentially ADSC differentiation.

Even though there was a strong correlation between CTCF and RNAPII binding at pear enriched TSSs, the correlation with RNA-seq data from the same cells was much lower, suggesting that there is significant regulation of ADSC gene expression that occurs after promoter engagement by RNAPII. It may indicate the presence of a “pausing” mechanism for these specific binding sites, meaning that CTCF triggers RNAPII binding at the TSS of target genes marked by active histone modifications, followed by a pause of transcription where RNAPII is waiting for additional signal(s) to continue transcription.

While studying the chromatin landscape of body shape-enriched CTCF binding sites, we identified the synchronous existence of functionally opposite histone marks (H3K4me3 and H3K27me3) at the TSSs of genes with body shape-enriched CTCF binding sites. This signature, defined as “bivalency”, has been shown to mark developmentally related genes [43]. After receiving developmental signals, the bivalent domain is converted by the removal of one mark that shifts the balance between gene activation and gene repression, which switches the gene transcription state from poised to on or off. Interestingly, this bivalent state was much more prevalent in CTCF sites in apple ADSC chromatin. The bivalent chromatin marks observed in the apple ADSCs could be associated with genes in ADSC chromatin that are primed for activation or repression by hormones, nutrients, or other signals during the differentiation process. In fact, the bivalent marked genes were significantly differentially expressed in ADSCs (undifferentiated) vs. adipocytes (differentiated), which is consistent with this model. The fact that the pear enriched CTCF binding sites were found primarily at the active TSSs of genes and were less associated with the presence of bivalent chromatin suggests that pear ADSCs may already be more committed to the adipogenic lineage than apple ADSCs. Further studies are needed. Importantly, identical observations were made in ABD and GF depots, revealing that this CTCF mechanism is body shape-specific rather than adipose tissue depot-dependent.

Our study revealed a unique epigenetic pattern around the promoter of *METRNL*. We observed simultaneous enrichment of CTCF and RNAPII binding at the TSS of *METRNL* in pear ADSCs, associated with the presence of a bivalent chromatin state upstream of the TSS. In parallel, we observed a decrease of expression of *METRNL* in pear subjects compared to that in apple subjects (data obtained from RNA-seq analysis). Altogether, these observations implied that CTCF and RNAPII binding at the gene TSS in pear ADSCs is not enough to turn on gene transcription. *METRNL* is an adipokine highly expressed in white adipose tissue [44], with comparable expression between adipocytes and stromal cells and is regulated by adipogenesis and obesity [45]. Its implication in adipogenesis is unclear, as it has been described as an inhibitor of adipocyte differentiation [46] as well as an activator of PPARG [38]. We found a positive correlation between *METRNL* and PPARG gene expression only in pear subjects, suggesting that *METRNL* could activate adipogenesis selectively in pear samples. The fact that *METRNL* expression is lower in pear ABD samples compared to that in apple ABD samples but is also associated with CTCF and RNAPII binding at its TSS implies that this gene is “poised” in pear subjects and ready to be activated. In this model, higher expression of *METRNL* in apple subjects not associated with PPARG expression indicates a different role in these subjects, potentially limiting ADSC differentiation into new adipocytes. Upon excess calorie consumption or hormonal signals, this restriction of differentiation would lead to adipocyte hypertrophy in the apple ABD depot along with an overall higher risk of developing metabolic syndrome. These results point out the importance of defining the body shape of the original donor in future investigations of the role of CTCF in human adipose tissue biology.

Importantly, CTCF expression was higher in ADSCs isolated from pear subjects only in the ABD depot, suggesting that CTCF expression is associated with reduced differentiation. A recent study similarly showed that CTCF expression in human ABD adipose tissue is inversely associated with BMI [17]. However, in this work, the relationship with body fat distribution was not explored, nor was the difference between apple and pear subjects. Thus, our new study provides a significant step forward and suggests that the decrease in CTCF expression associated with BMI reflects a mechanism of protection utilized by stem cells to limit the adipogenic expansion of adipose tissue.

## 5. Conclusions

Taken together, our data support a model whereby CTCF regulates differential fat distribution in women with upper vs. lower body obesity by: (1) binding more genes in pear-shaped ADSCs, (2) targeting the active promoter region of those same genes, and (3) potentially recruiting or influencing RNAPII activity at these genomic regions. This work expands our previously reported results on the ABD and GF adipose tissue-specific transcriptional signature associated with body shape and provides evidence and an underlying mechanism for the epigenetic control of fat distribution in women.

## Figures and Tables

**Figure 1 cells-13-00086-f001:**
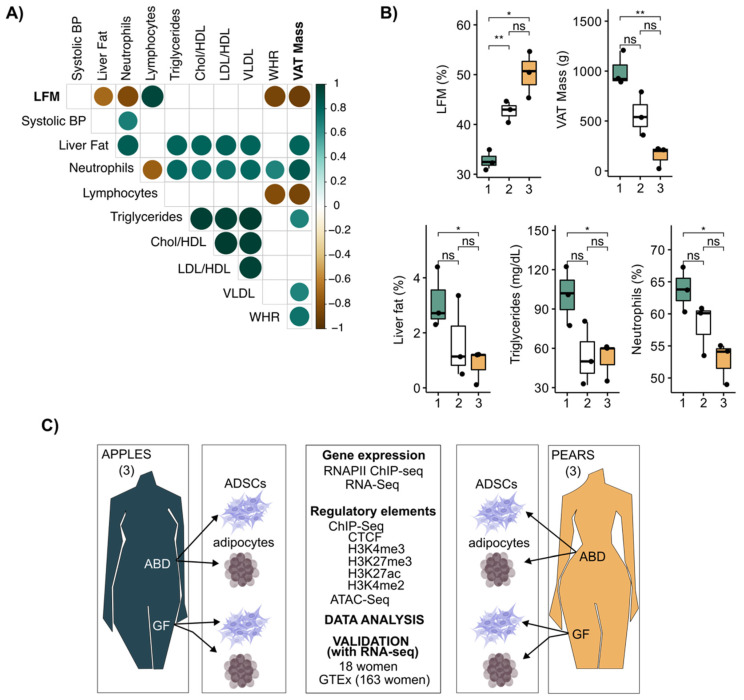
LFM and VAT mass defined the body shape (apples and pears) in women. (**A**) Correlation plot showing correlation coefficients between clinical parameters for *n* = 9 women. Spearman correlation, rho > |0.4|, *p*-value < 0.05. (**B**) Boxplots showing differences in clinical parameters between each group separated by LFM. *t*-test, ** *p*-value < 0.01, * *p*-value < 0.05, ns = not significant. (**C**) Schematic illustration of the study design. ABD: abdominal; GF: gluteofemoral; ADSCs: adipose-derived stem cells.

**Figure 2 cells-13-00086-f002:**
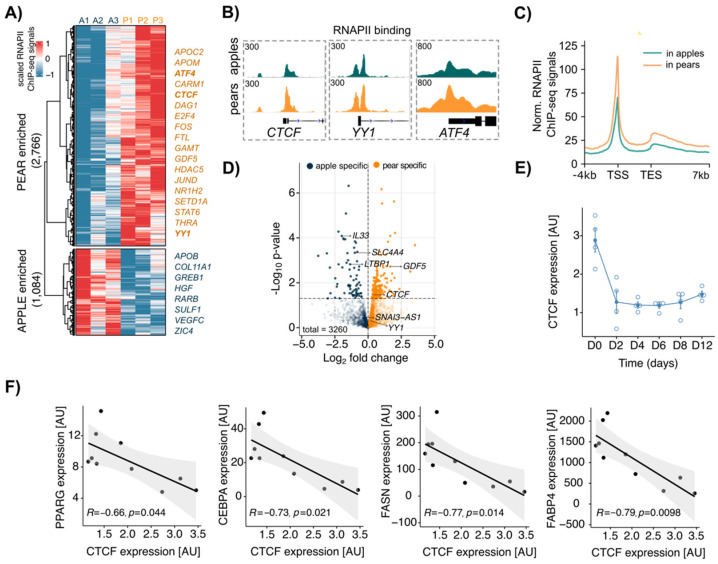
Differential RNAPII binding at transcription start site (TSS) between apple vs. pear ABD ADSCs showed body shape-specific transcriptional signatures in the SAT depot. (**A**) Clustered heatmap showing normalized values of 3850 body shape-specific RNAPII bindings in each subject in ABD ADSCs. A = apple, P = pear. Scaled RNAPII ChIP-seq signals were obtained at the TSS (±2 kbp). (**B**) IGV screenshot depicting RNAPII bindings at CTCF, YY1, and ATF4 TSS in apple and pear samples (average signal for *n* = 3 subjects per group). (**C**) Metagene profile representing normalized RNAPII ChIP-seq signals in apple and pear samples at genes with body shape-specific RNAPII binding. (**D**) Volcano plot showing differentially expressed genes between apple and pear samples based on RNA-seq data. The genes with body shape-enriched RNAPII binding are represented in indicated colors. (**E**) CTCF expression measured by RT-qPCR during differentiation. Average of *n* = 4 ADSCs. D: day. (**F**) Plots showing linear regression between expression of CTCF and expression of PPARG, CEBPA, FASN, and FABP4 at the end of differentiation of ABD ADSCs (D12). *n* = 10 subjects.

**Figure 3 cells-13-00086-f003:**
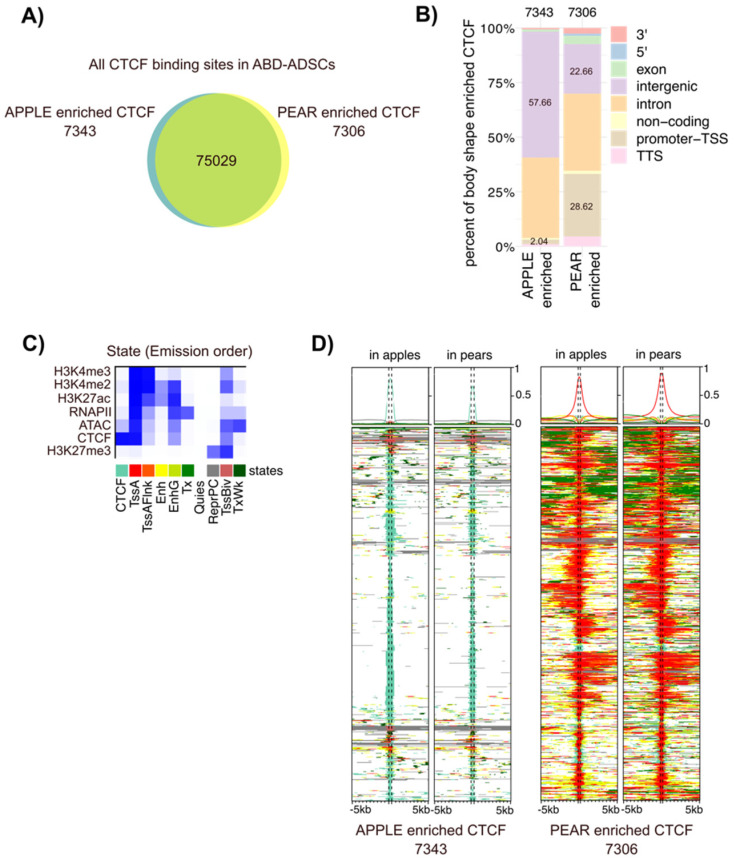
Apple and pear enriched CTCF binding sites target different genomic regions. (**A**) Proportional Venn diagrams showing the number of genes with pear enriched (yellow), apple enriched (blue), and common CTCF binding sites (intersection) in ABD ADSCs. (**B**) Bar graph representing the genomic distribution of the body shape-enriched CTCF binding sites in ABD ADSCs. (**C**) Heatmap of the emission parameters in which each column corresponds to a different state and each row corresponds to a different mark for four histone modifications (H3K4me2, H3K4me3, H3K27me3, and H3K27ac), ATAC-seq, RNAPII, and CTCF. (**D**) ChromHMM heat map representation showing signals around the apple (left side) and pear (right side) enriched CTCF binding sites in apple and pear samples (*n* = 6). The colors referred to the different states in (**C**).

**Figure 4 cells-13-00086-f004:**
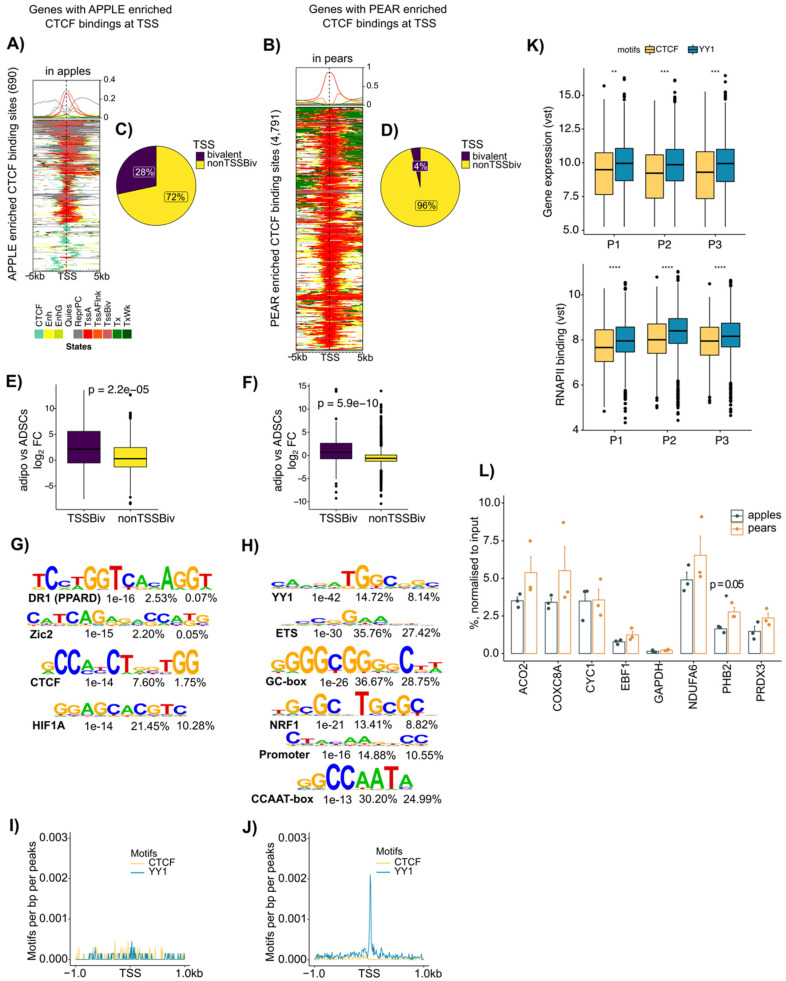
Apple enriched CTCF binding is associated with bivalent TSS, and pear enriched CTCF binding is associated with active TSS and YY1 motif in ABD ADSCs. (**A**) ChromHMM heat map representation showing signals around the 690 apple enriched CTCF binding sites in apple samples (*n* = 3). (**B**) ChromHMM heat map representation showing signals around the 4791 pear enriched CTCF binding sites in pear samples (*n* = 3). (**C**,**D**) Pie chart showing the percentage of apple-specific CTCF binding genes with bivalent TSSs and the percentage of pear-specific CTCF binding genes with bivalent TSSs. (**E**,**F**) Boxplots showing fold change (FC) of expression between ADSCs versus adipocytes at genes with apple enriched CTCF binding with or without bivalent TSSs and at genes with pear enriched CTCF binding with or without bivalent TSSs. Unpaired two-samples *t*-test. (**G**,**H**) Motif enrichment analysis for apple and pear enriched CTCF binding peaks. Enriched motif matrices are presented along with the *p*-value. The percentages of each motif found in the target and background genomic regions are indicated. (**I**,**J**) Histograms showing motif densities at apple and pear enriched CTCF binding. CTCF motif enrichment is represented in yellow, and YY1 motif enrichment is represented in blue. (**K**) Boxplots showing gene expression differences between genes with YY1 (blue) and CTCF (yellow) motifs at their TSS. Variance stabilizing transformation (vst) of counts from each pear subject are shown by using RNAPII ChIP-seq (**left**) and RNA-seq (**right**). ** *p* < 0.01, *** *p* < 0.001, **** *p* < 0.001. (**L**) Bar graph showing YY1 binding at selected genes with pear enriched CTCF binding sites from three apple and three pear ABD ADSCs (*n* = 6). IgG was used as the control. Unpaired *t*-test was used.

**Figure 5 cells-13-00086-f005:**
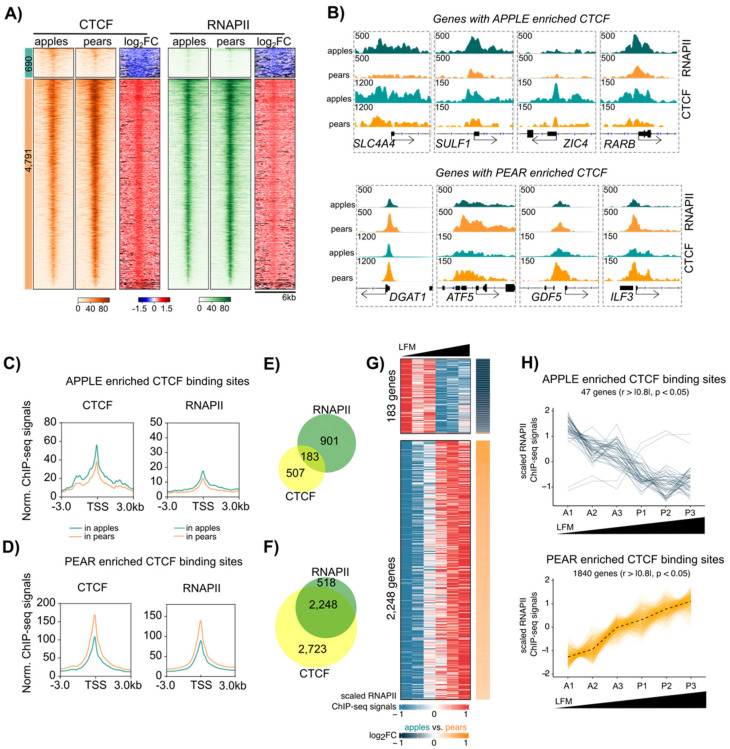
Association between body shape enriched CTCF and RNAPII binding at the transcription start site (TSS) in ABD ADSCs. (**A**) Heatmap showing CTCF (orange) and RNAPII (green) enrichment in apple and pear samples at apple (left heat maps) and pear (right heat maps)-specific CTCF binding sites within ±3 kb frame of TSS. Log_2_ FC heatmaps represent the ratio of CTCF or RNAPII signals between apple and pear samples. (**B**) IGV screenshots showing genes with apple-specific (*SLC4A4*, *SULF1*, *ZIC4*, and *RARB*) and pear-specific (*DGAT1*, *ATF5*, *GDF5*, and *ILF3*) RNAPII and CTCF binding at their TSS. Teal plots represent the average of three apple samples, and orange plots represent the average of three pear samples. (**C**) Metagene profile representing normalized CTCF and RNAPII ChIP-seq signals in apple (teal) and pear (orange) samples around gene TSSs with apple enriched CTCF binding. (**D**) Metagene profile representing normalized CTCF and RNAPII ChIP-seq signals in apple (teal) and pear (orange) samples around gene TSSs with pear enriched CTCF binding. (**E**) Proportional Venn diagrams showing the number of genes with only RNAPII (green), only CTCF (yellow), or simultaneously RNAPII and CTCF (intersection) apple enriched binding sites at their TSS. (**F**) Proportional Venn diagrams showing number of genes with only RNAPII (green), only CTCF (yellow), or simultaneously RNAPII and CTCF (intersection) pear enriched binding sites at their TSS. (**G**) Heatmaps showing individual RNAPII ChIP-seq signal at the TSSs of 183 and 2248 genes with apple and pear CTCF/RNAPII enriched binding sites, respectively. The subjects are sorted by LFM. Log_2_ FC heatmaps (right side) represent the ratio of RNAPII signals between apple and pear samples; orange indicates a higher signal in pear samples, and blue indicates a higher signal in apple samples. (**H**) Lineplots showing scaled RNAPII ChIP-seq signals at gene TSSs of apple (top) or pear (bottom) enriched CTCF/RNAPII binding sites in each subject ordered by LFM. *n* = 6 subjects. Only genes presenting a correlation between scaled RNAPII ChIP-seq signals and LFM are shown (Spearman, rho > |0.8|, *p*-value < 0.05).

**Figure 6 cells-13-00086-f006:**
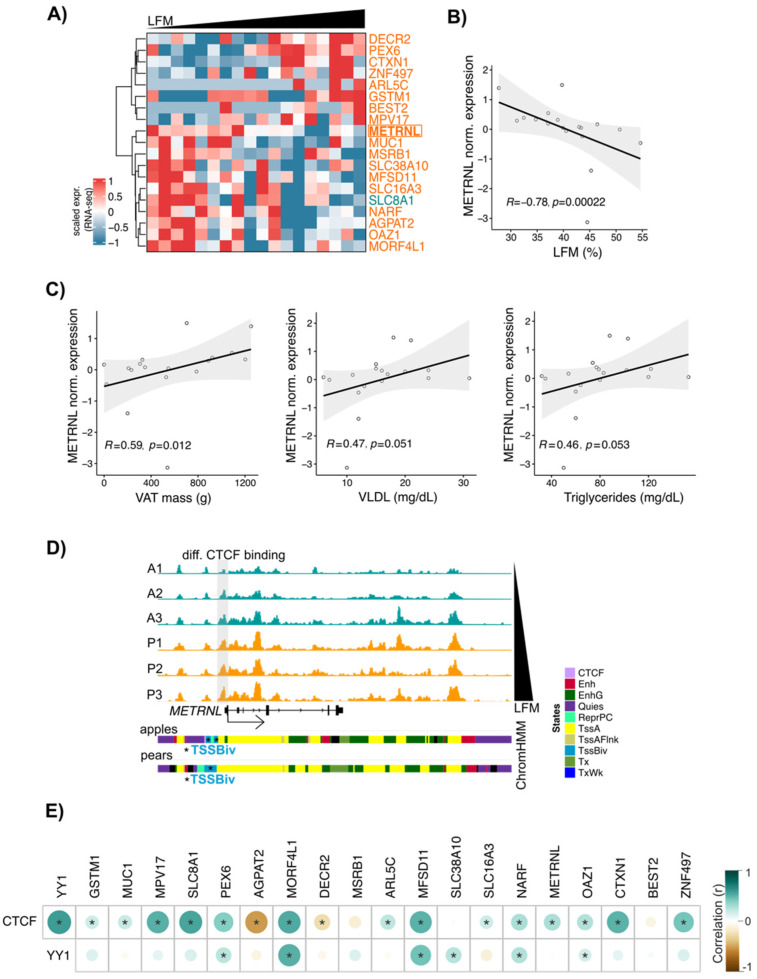
*METRNL*, a potential CTCF target gene, correlates with LFM. (**A**) Heatmap showing the scaled gene expression (RNA-seq) in ABD ADSCs of genes with body shape-enriched CTCF at their TSS. *n* = 18 subjects ranked by LFM. Only genes having a correlation between scaled gene expression and LFM are shown (Spearman, *p*-value < 0.05). The gene colored in teal have apple enriched CTCF bindings at its TSS. Genes colored in orange have pear enriched CTCF bindings at their TSS. (**B**) Scatterplot showing correlation between LFM and *METRNL* gene expression. *n* = 18 ABD ADSCs. Spearman correlation. (**C**) Scatterplots showing correlations between *METRNL* gene expression and VAT mass, circulating VLDL, and triglyceride levels. *n* = 18 ABD ADSCs. Spearman correlation. (**D**) IGV screenshot showing increase of CTCF signals at *METRNL* TSS in parallel to increased LFM. ChromHMM representation showing the presence of bivalent TSS state at the TSS of *METRNL*. (**E**) Corrplot showing correlations between CTCF and YY1 gene expression and CTCF target gene expression based on (**A**). RNA-seq data derived from GTEx database for *n* = 163 women adipose tissue biopsies. * *p* < 0.05, Spearman correlation.

## Data Availability

All sequencing data have been deposited to the NCBI GEO database (http://www.ncbi.nlm.nih.gov/geo/ (accessed on 30 June 2023)) under accession numbers GSE224770 (ChIP-seq and ATAC-seq), GSE193812 (RNA-seq from 18 ADSCs), and GSE143450 (RNA-seq from adipocytes). GTEx data were obtained from pht002742.v9.p2 version.

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
