# Peer review of "Transcriptional Control of Subcutaneous Adipose Tissue by the Transcription Factor CTCF Modulates Heterogeneity in Fat Distribution in Women"

_cells, 2023, doi:10.3390/cells13010086_

Round 1
Reviewer 1 Report
Comments and Suggestions for Authors
The present manuscript expands previously reported results on the ABD and GF adipose tissues specific transcriptional signatures associated with body shape along with epigenetic patterns. The authors, trying to understand the mechanism underlining the different behaviour of adipose tissue in apple- and pear-shaped women, addressed the study of epigenetic features of ADSC derived from adipose tissue of these two anatomical categories of women. The authors found different CTCF binding sites together with other epigenetic features in ADSC derived from apple- and pear-shaped patients and this gave rise to a biological model that can explain the differential regulation of METRNL and PPARG genes as well as the different capability of adipocyte differentiation in the two categories of samples. The study is well written and well discussed. The subject is highly topical and addressed with innovative methodologies. The most limitation of this study is the limited number of patients recruited in this study. However, I consider that the data presented contain enough novelty and therefore the manuscript deserves to be published on “Cell”. There are some minor points to be addressed that needs to be fixed.
Minor points:
-In the "materials and methods" section (2.1 paragraph) the authors stated that ADSCs derived from gluteo-femoral biopsy, but in the results the authors claimed that they studied ADSC from GF and ABD (3.1 paragraph). A clarification is needed.
-In my opnion the captions of supplementary figures and tables should be inserted in the same files that contain each supplementary figure and table.
-A paragraph with the list of abbreviations would improve the clarity of the text.
Reviewer 2 Report
Comments and Suggestions for Authors
The manuscript by Erdos et al examines the role of CTCF in adipose tissue. This studies used human cells to examine differences between fat distributions. They observed that CTCF regulation is more substantial in pear shaped individuals. The CTCF binding was also in different gene regions when comparing apple and pear shaped donors. This is very well written paper. The introduction is particularly informative. The layout of the paper could be improved such at the figures and matching figure legends are on the same page. In figure legend 2, 4, and 5, the figure titles should be in bold as in the other figures. Also, there should be consistency in the bolding of figure legend sections (A, B, etc.).
Overall, this manuscript contains a lot of novel data that underscores the complexity of human adipose tissue.
Reviewer 3 Report
Comments and Suggestions for Authors
1. Gender dependence: The title and abstract of the manuscript do not mention the gender of the pear and apple donors. Therefore, it is unclear whether the authors' results for CTCF, YY1, METRNL, etc. differ by gender and whether these differences apply to general conclusions about human adipose tissue and obesity.
2. Organization of Figure Data: Figures 2, 4, and 5 present data derived from the ventral depot, whereas Figures 3 and 6 present data derived from the ABD-ADSC. Make sure this data structure supports the author's claims. The transcription factor CTCF regulates the heterogeneity of fat distribution through transcriptional regulation in human subcutaneous adipose tissue, which can only be determined by considering the overall context and results of the paper.
3. Highlighted in Figure 6A: If SLC8A1 is highlighted instead of METRNL in Figure 6A, this may mean a mistake.
4. Please check double spaces on lines 571 and 580.
